# From Silent to Life-Threatening: Giant Left Atrial Myxoma Presenting with Acute Pulmonary Edema—A Case Report

**DOI:** 10.3390/reports8030170

**Published:** 2025-09-05

**Authors:** Ciprian Nicușor Dima, Marinela-Adela Scuturoiu, Iulia-Raluca Munteanu, Alis Liliana Carmen Dema, Horea Bogdan Feier

**Affiliations:** 1Department VI Cardiology-Cardiovascular Surgery Clinic, “Victor Babes” University of Medicine and Pharmacy, 300041 Timisoara, Romania; dima.ciprian@umft.ro (C.N.D.); horea.feier@umft.ro (H.B.F.); 2Institute for Cardiovascular Diseases of Timisoara, Clinic of Cardiovascular Surgery, Gheorghe Adam Street, No. 13A, 300310 Timisoara, Romania; 3Doctoral School Medicine-Pharmacy, “Victor Babes” University of Medicine and Pharmacy, 300041 Timisoara, Romania; 4NEUROPSY-COG Center for Cognitive Research in Neuropsychiatric Pathology, Department of Neuroscience, “Victor Babes” University of Medicine and Pharmacy, 300041 Timisoara, Romania; 5Advanced Research Center, Institute for Cardiovascular Diseases, 300310 Timisoara, Romania; 6Department of Pathology, “Victor Babes” University of Medicine and Pharmacy, 300041 Timisoara, Romania; dema.alis@umft.ro

**Keywords:** left atrial myxoma, cardiac tumors, pulmonary edema, mitral valve obstruction

## Abstract

**Background and Clinical Significance:** Cardiac myxomas, though typically benign and asymptomatic, can rarely present with acute cardiovascular compromise. We report a case of a left atrial myxoma presenting as acute pulmonary edema in a patient with prior normal cardiac imaging. **Case Presentation:** A 55-year-old male, with a history of thrombolyzed myocardial infarction and normal coronary angiography and echocardiography five years earlier, was admitted with acute dyspnea and pulmonary edema. Bedside transthoracic echocardiography (TTE) revealed a left atrial mass causing severe mitral inflow obstruction. Emergency surgical excision was performed, and the mass was submitted for histopathological analysis. **Discussion:** Histology confirmed cardiac myxoma. The procedure and recovery were uneventful, and follow-up at one month confirmed no recurrence. **Conclusions:** This case illustrates the potential for cardiac myxoma to manifest suddenly with life-threatening symptoms, even after previously normal investigations. Echocardiography remains pivotal in diagnosing intracardiac masses and guiding timely intervention.

## 1. Introduction and Clinical Significance

Cardiac myxomas are the most common primary cardiac tumors, representing 50–85% of cases, with about 75% localized in the left atrium, usually at the interatrial septum near the fossa ovalis. Epidemiological studies estimate an incidence of approximately 0.5 cases per million population per year. They are most often diagnosed between the ages of 30 and 60 years, with a slight female predominance (about 2:1) [1,2]. Although benign and often slow-growing, they can occasionally present with sudden and severe manifestations [3].

Recent reviews and multicenter studies have provided a more comprehensive understanding of cardiac myxomas, including their clinical spectrum, risk factors for embolic events, and surgical outcomes [1,4,5]. These reports emphasize that while most myxomas are sporadic and benign, their unpredictable growth and potential for catastrophic complications justify early recognition and prompt surgical excision. In many cases, myxomas remain asymptomatic for prolonged periods, only becoming clinically evident when they reach a size sufficient to cause hemodynamic compromise, embolic events, or systemic manifestations [6]

The clinical presentation of cardiac myxomas is heterogeneous and can be broadly categorized into three symptom clusters: obstructive, embolic, and constitutional. Obstructive manifestations are the most common, reported in 50–60% of cases, followed by embolic events in 30–40% and constitutional symptoms in 10–20% [1,4,7,8]. Obstructive symptoms arise from interference with intracardiac blood flow, leading to dyspnea, orthopnea, syncope, or signs mimicking mitral valve disease. Embolic events may result from tumor fragmentation or thrombus formation on the tumor surface, leading to cerebrovascular accidents, peripheral arterial occlusion, or visceral infarctions [1,7,8,9]. Rarely, cardiac myxomas may cause an abrupt clinical presentation with acute pulmonary edema, resulting from sudden and significant obstruction of the mitral valve by a large and mobile mass, precipitating a critical hemodynamic state.

From a diagnostic standpoint, the current clinical recommendations emphasize the pivotal role of echocardiography as the initial imaging modality of choice when a cardiac mass is suspected, given its widespread availability, non-invasive nature, and high diagnostic accuracy. Transthoracic echocardiography provides valuable information regarding the size, mobility, location, and hemodynamic impact of the mass. In cases where image quality is suboptimal or detailed anatomical assessment is required, transesophageal echocardiography (TEE) is preferred due to its superior resolution, particularly for left atrial tumors. Despite these advantages, echocardiography has certain limitations. Small tumors may occasionally be missed, image quality is operator-dependent, and in some cases, it may be challenging to differentiate a myxoma from other intracardiac masses such as thrombus. Cardiac MRI or CT can therefore provide complementary information when further tissue characterization is required. Cardiac magnetic resonance imaging (MRI) and computed tomography (CT) may offer complementary data, especially when differential diagnosis with other cardiac masses such as thrombus, papillary fibroelastoma, or malignant neoplasms is necessary [4,5,10].

According to current guidelines, once a cardiac myxoma is diagnosed or highly suspected, early surgical resection is strongly recommended, regardless of symptom severity, due to the risk of sudden obstruction or embolization. International recommendations emphasize that surgical excision should not be delayed, even in asymptomatic patients, as conservative management carries a significant risk of fatal complications [11,12].

## 2. Case Presentation

A 55-year-old male with no known comorbidities presented to the emergency department with acute-onset dyspnea at rest, orthopnea, and palpitations that had progressively worsened over the preceding 24 h. Notably, five years prior to the current presentation, the patient suffered an episode of acute myocardial infarction, which was managed with thrombolytic therapy. During that event, he developed a cardiorespiratory arrest, requiring advanced cardiac life support, with successful resuscitation. Following stabilization, a comprehensive cardiovascular assessment was performed, including transthoracic echocardiography (TTE) and coronary angiography, both of which demonstrated normal findings, with no evidence of intracardiac masses, structural cardiac abnormalities, or significant coronary artery disease. Since that episode, the patient had been maintained on long-term aspirin therapy, with no recurrent cardiac symptoms reported until the current presentation.

On arrival, the patient was in acute respiratory distress, with tachypnea (30 breaths per minute), tachycardia (108 beats per minute), and blood pressure of 180/100 mmHg. His oxygen saturation was 87% on room air, with modest improvement following high-flow oxygen therapy. On examination, auscultation revealed a soft mid-diastolic murmur at the cardiac apex and bilateral coarse crackles throughout both lung fields. There was no jugular venous distension, peripheral edema, or clinical signs of embolic phenomena. Initial laboratory tests, including cardiac biomarkers, were within normal ranges.

Due to the severity of the presentation and the unclear etiology, an urgent bedside transthoracic echocardiogram (TTE) was performed. This examination revealed a large, mobile, pedunculated mass measuring approximately 3.7 × 4.5 cm, located in the left atrium and attached to the interatrial septum at the level of the fossa ovalis. The mass was prolapsing through the mitral valve during diastole, causing significant dynamic obstruction of left ventricular inflow, with a mean transmitral gradient of 14 mmHg (Figure 1). This finding was consistent with severe functional mitral stenosis and provided a clear explanation for the patient’s acute pulmonary edema and hemodynamic instability. No transesophageal echocardiography was deemed necessary, as the transthoracic study provided sufficient diagnostic information to confirm the presence of the obstructive intracardiac mass and to guide urgent surgical decision-making.

Given the acute presentation and echocardiographic evidence of severe mitral inflow obstruction caused by the large left atrial mass, the clinical team did not pursue repeat coronary angiography. There were no signs or symptoms suggestive of myocardial ischemia, ECG changes were nonspecific, and cardiac biomarkers were within normal range. Therefore, other differential diagnoses such as spontaneous coronary artery dissection or vasospasm were considered unlikely in this context.

Given the critical obstructive physiology and life-threatening presentation, the patient underwent urgent surgical excision of the mass via median sternotomy under cardiopulmonary bypass (Figure 2). Upon opening the left atrium, a gelatinous mass was identified, attached to the interatrial septum via a narrow pedicle. The tumor measured approximately 3.7 × 4.5 cm and exhibited a lobulated, irregular surface (Figure 3). These findings accounted for the severe dynamic obstruction of the mitral orifice observed preoperatively and underscore the capacity of left atrial myxomas to precipitate acute, life-threatening obstructive disease. Intraoperatively, the mass was friable and gelatinous, with focal areas of increased density suggestive of fibrous components, consistent with a mixed morphological pattern. Its marked mobility and fragile texture raised concern for embolic potential, characteristic of the macroscopic features of cardiac myxomas. The tumor was completely excised at its point of attachment, without requiring interatrial septal resection or reconstruction.

The surgical procedure was uneventful, with no intraoperative or postoperative complications. The postoperative course was favorable, with resolution of pulmonary symptoms and no recurrence of arrhythmias or hemodynamic instability. Transthoracic echocardiography performed prior to discharge confirmed complete excision of the mass, with no residual obstruction or valvular dysfunction. The patient was discharged in stable condition on the seventh postoperative day.

The excised mass was submitted for histopathological examination, which confirmed the diagnosis of cardiac myxoma.

Histopathological examination confirmed the diagnosis of cardiac myxoma, showing tumor cells arranged in nests and cords within a myxoid stroma, with fibrino-hematic deposits on the surface (Figure 4 and Figure 5).

Preoperative imaging is essential. Transthoracic ultrasound is the first-line evaluation method, but transesophageal ultrasound is superior, useful when transthoracic is inconclusive, and can be sensitive in preoperative planning.

Cardiac MRI is the best method for characterizing tissues and differentiating thrombi from other masses, but in our emergency case, the timing and logistics did not permit cardiac MRI.

At 1-month follow-up, transthoracic echocardiography showed no evidence of residual or recurrent mass, with normal left atrial and mitral valve function.

## 3. Discussion

This case illustrates several clinically relevant aspects regarding the unpredictable natural history and potentially catastrophic clinical presentation of cardiac myxomas. Although traditionally regarded as slow-growing, benign neoplasms, our case demonstrates that cardiac myxomas can remain clinically silent for extended periods and then present abruptly with severe, life-threatening hemodynamic compromise. Five years earlier, the patient had an acute myocardial infarction, with normal coronary angiography and echocardiography at that time. There was no evidence of a cardiac mass during that episode. Therefore, the present obstructive myxoma should be regarded as a new and independent finding, highlighting the unpredictable natural history of these tumors, which may remain silent until reaching a critical size.

Several case reports have documented cardiac myxomas presenting with acute pulmonary edema or mimicking acute coronary syndromes [13,14,15]. A distinguishing feature of this case is the five-year interval between a previously normal echocardiographic evaluation and the subsequent abrupt onset of life-threatening obstruction. Although indolent growth of cardiac myxomas has been documented, reports of sudden clinical deterioration in the absence of preceding symptoms or structural abnormalities are rare. This observation underscores the unpredictable biological behavior of these tumors and raises the possibility that subtle manifestations were overlooked or, alternatively, that the lesion developed de novo.

Although cardiac myxomas are most often associated with embolic or constitutional symptoms, their obstructive presentation is rare and may manifest as acute pulmonary edema or orthostatic syncope when the tumor intermittently occludes the mitral orifice. In our case, the sudden pulmonary edema was due to a large left atrial mass prolapsing through the mitral valve, producing severe functional obstruction. This physiology can closely mimic rheumatic mitral stenosis, as both conditions may present with dyspnea, pulmonary congestion, and a diastolic murmur. Echocardiography was decisive, revealing a mobile mass rather than a fixed valvular lesion, underscoring the importance of considering myxoma in the differential diagnosis of apparent mitral stenosis, particularly in acute or atypical cases.

The growth kinetics of cardiac myxomas remain poorly defined. Case reports and small series suggest rates from a few millimeters to over a centimeter per year, but these estimates are speculative given the rarity of serial imaging [6]. Furthermore, inter-patient variability is significant, and factors influencing tumor growth—such as genetic predisposition, local hemodynamic factors, or inflammatory mediators—are yet to be fully elucidated [12]. Our case underscores the potential value of tailored surveillance in selected patients, especially those with prior unexplained cardiac events despite initially normal imaging.

Based on these literature estimates, a myxoma measuring 3.7 × 4.5 cm could potentially develop over 3–5 years, assuming a linear or slightly accelerated growth pattern. This aligns with the five-year interval observed in our case, although interindividual variability and non-linear growth cannot be excluded.

Current guidelines offer no consensus on routine screening or follow-up imaging in asymptomatic patients with prior cardiac events, and our case highlights this gap in practice [13]. Larger studies and registries are needed to better define the natural history of myxomas and to develop evidence-based protocols for risk stratification, surveillance, and early detection—particularly in atypical or higher-risk patients. Clinically, this case was challenging because of its acute and atypical presentation. The patient arrived with severe pulmonary edema and hemodynamic instability, initially suggestive of acute coronary syndrome—a frequent pitfall in emergency settings where sudden respiratory failure and collapse occur. Given his history of myocardial infarction, immediate coronary assessment was prioritized, but both TEE and ECG showed no significant changes in coronary assessment, which, however, revealed no significant changes on TEE and ECG.

The differential diagnosis at the time of initial imaging included left atrial thrombus, papillary fibroelastoma, and, less likely, metastatic tumors. However, the echocardiographic characteristics—specifically the large size, location at the interatrial septum, pedunculated attachment, and prolapse through the mitral valve during diastole—were highly suggestive of myxoma. The absence of atrial fibrillation, valvular disease, or known malignancy further reduced the likelihood of thrombus or secondary masses.

The diagnostic ambiguity illustrates the difficulty of evaluating acute cardiogenic pulmonary edema. In such settings, point-of-care transthoracic echocardiography (TTE) is essential, offering rapid, non-invasive assessment of cardiac structure and function [14]. In our patient, TTE was decisive, identifying a large left atrial mass prolapsing into the mitral valve and explaining both pulmonary edema and hemodynamic compromise. This case emphasizes the value of TTE as the first-line imaging tool in unstable patients, where transesophageal echocardiography (TEE) may be impractical or contraindicated. Although TEE provides superior resolution, the urgency of the scenario and adequacy of TTE findings made further imaging unnecessary. Early bedside echocardiography remains crucial in undifferentiated acute heart failure, allowing rapid recognition of reversible mechanical causes and timely surgical intervention [4,5,10].

The management of this case highlights the critical importance of timely surgical intervention in patients presenting with obstructive cardiac masses causing hemodynamic instability. Once the diagnosis of a left atrial mass was established through bedside echocardiography, the decision to proceed with urgent surgical excision was essential to prevent further hemodynamic deterioration, embolic complications, or recurrent pulmonary edema. The surgical approach was straightforward, involving complete excision of the mass via standard median sternotomy under cardiopulmonary bypass. Given the tumor’s attachment by a narrow pedicle, resection was achieved without requiring interatrial septal reconstruction, contributing to a favorable surgical outcome. Large case series and meta-analyses confirm that surgical excision is associated with excellent outcomes, with recurrence rates below 5% in sporadic cases, but higher in familial or syndromic forms [4,15]

The patient’s postoperative course was uneventful, with rapid clinical recovery, and early echocardiographic follow-up confirmed the absence of residual or recurrent mass. This reinforces existing literature indicating that surgical excision of myxomas, when performed promptly and completely, is associated with excellent short- and long-term outcomes. Nonetheless, this case also reflects the necessity for heightened clinical awareness and rapid decision-making, particularly in acute presentations where delayed intervention could result in irreversible complications [2]. Multidisciplinary coordination and early surgical consultation are essential to optimize outcomes in such high-risk scenarios [15,16].

This case also highlights limitations relevant to future practice. Despite a major cardiac event five years earlier, including myocardial infarction with cardiorespiratory arrest, no follow-up echocardiography was scheduled. Surveillance might have revealed the myxoma at an earlier, asymptomatic stage. At that time, imaging and clinical findings showed no evidence of an intracardiac mass—echocardiography was normal, and there were no murmurs or embolic signs. In retrospect, if suspicion had existed, transesophageal echocardiography (TEE) or cardiac CT would have offered greater sensitivity for early or small lesions. This underscores the diagnostic value of advanced imaging in selected high-risk or ambiguous cases.

Following discharge, the patient was enrolled in a structured follow-up protocol including clinical and transthoracic echocardiographic evaluation at 1, 6, and 12 months. Long-term annual surveillance was also recommended due to the known, albeit low, risk of recurrence, particularly in sporadic cases.

Moreover, while the emergent clinical scenario appropriately prioritized rapid bedside transthoracic echocardiography (TTE), the absence of preoperative transesophageal echocardiography (TEE) may be viewed as a potential limitation. Although TTE provided sufficient diagnostic information to guide urgent surgical management, TEE could have offered superior visualization of the tumor’s attachment, morphology, and relationship to adjacent structures, potentially refining surgical planning [4,5]. This consideration reinforces the importance of balancing the urgency of intervention with the potential added value of more detailed imaging modalities when the patient’s condition allows.

## 4. Conclusions

This case reinforces several critical clinical messages that may inform both acute management and long-term strategies for patients presenting with unexplained cardiogenic pulmonary edema or hemodynamic collapse. First, obstructive cardiac tumors, particularly left atrial myxomas, should always be considered in the differential diagnosis of acute pulmonary edema. Maintaining a broad diagnostic perspective in such scenarios is essential to avoid delays in identifying reversible mechanical causes.

Second, point-of-care transthoracic echocardiography (TTE) should be integrated early in the evaluation of unexplained acute heart failure or hemodynamic instability, as it offers rapid, non-invasive, and readily available diagnostic insight. In this case, TTE was decisive in detecting the obstructive left atrial mass and guiding timely surgical referral. While transesophageal echocardiography (TEE) can provide additional anatomical detail, in critically ill patients, immediate management decisions should not be deferred when TTE yields sufficient diagnostic information.

Third, timely surgical excision remains the cornerstone of treatment for cardiac myxomas, with generally excellent outcomes when performed urgently. Delays in surgical intervention can expose patients to heightened risks of embolization, recurrent pulmonary edema, or sudden cardiovascular collapse.

Finally, this case also prompts consideration of individualized imaging follow-up strategies in patients with prior unexplained cardiac events, even when initial diagnostic workup, including echocardiography, is normal. Although current guidelines do not mandate routine surveillance in such cases, tailored approaches may be appropriate in selected high-risk individuals. Further research is needed to better define at-risk populations and to establish evidence-based protocols for surveillance and early detection of cardiac tumors.

## Figures and Tables

**Figure 1 reports-08-00170-f001:**
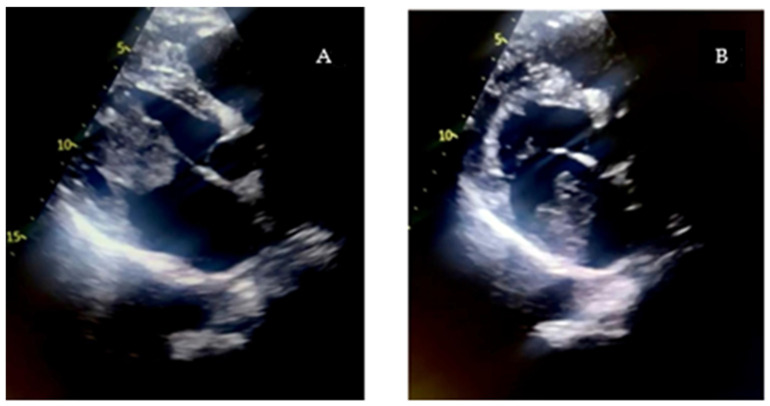
TTE Showing Cardiac Mass During Systole (**A**) and Diastole (**B**).

**Figure 2 reports-08-00170-f002:**
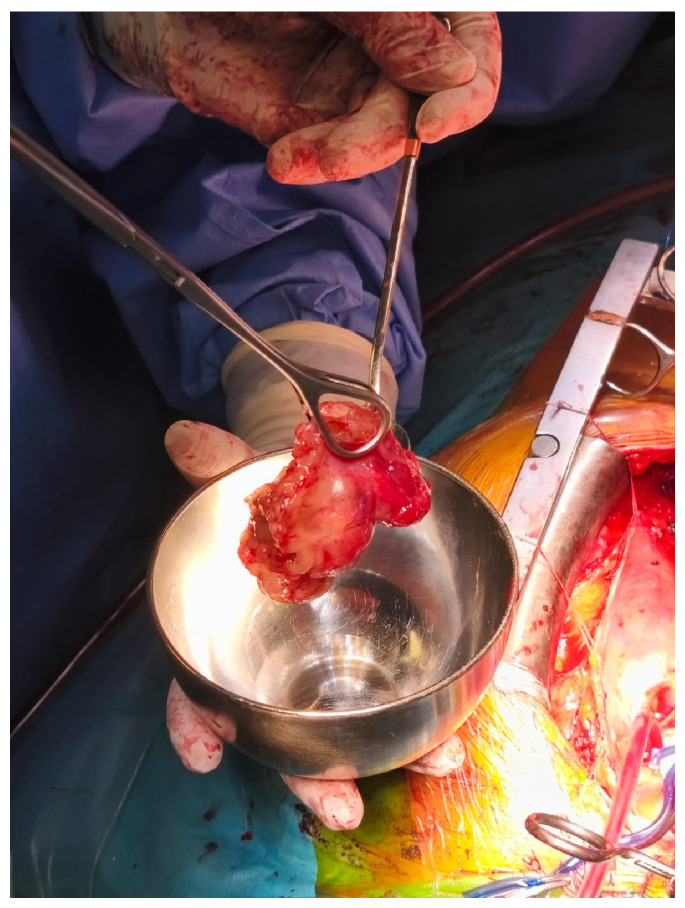
Intraoperative view of the left atrial myxoma during surgical excision.

**Figure 3 reports-08-00170-f003:**
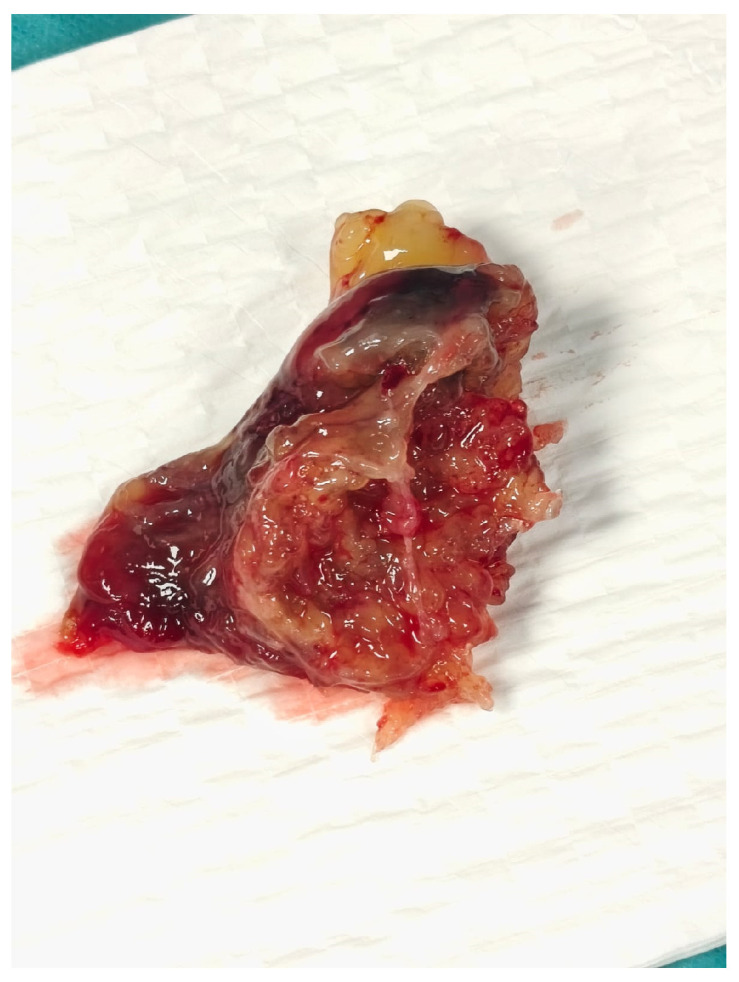
Gross specimen of the excised left atrial myxoma, measuring approximately 3.7 × 4.5 cm, with a lobulated and irregular surface.

**Figure 4 reports-08-00170-f004:**
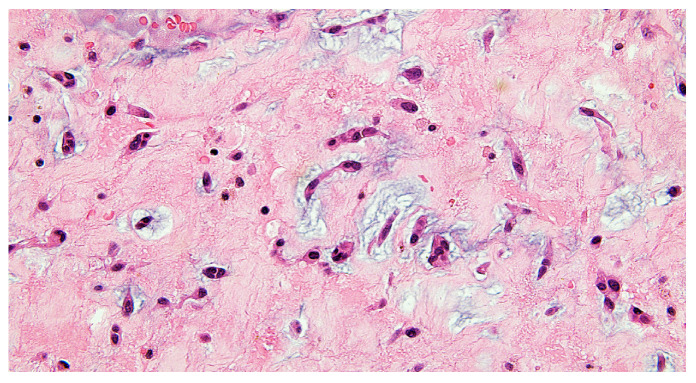
Intra-atrial myxoma—tumor cells arranged in a myxoid stroma, fibrino-hematic deposit on the tumor surface. HEx100.

**Figure 5 reports-08-00170-f005:**
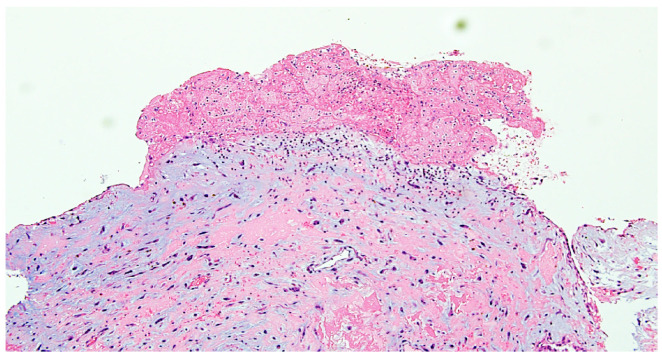
Intra-atrial myxoma —detail with tumor cells arranged in nests and short cords. HEx400.

## Data Availability

The original contributions presented in this study are included in the article. Further inquiries can be directed to the corresponding authors.

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
