# Peer review of "From Silent to Life-Threatening: Giant Left Atrial Myxoma Presenting with Acute Pulmonary Edema—A Case Report"

_reports, 2025, doi:10.3390/reports8030170_

Round 1
Reviewer 1 Report
Comments and Suggestions for Authors
The manuscript illustrates a classic case of obstructive myxoma.
It can greatly benefit from being much more concise, and from focused highlighting of:
1. the rare obstructive presentation of myxoma, with either orthostatic syncope or with pulmonary edema
2. the classic differential diagnosis with mitral stenosis, particularly relevant in this case given the presence of a diastolic murmur
The link with the previous presentation with acute coronary syndrome is far fetched with no evidence for a myxoma at that time. It is purely speculative and serves only as a reminder that not all infarcts are atherosclerotic in their etiology.
The echocardiographic images are good, the histopathology illustrations are excessive and do not serve the goal of the manuscript.
A shorter manuscript focused on the obstructive nature of this myxoma would read much easier
Author Response
We sincerely thank the reviewer for the constructive feedback. We carefully revised the manuscript in line with the suggestions, and all modifications have been highlighted in yellow for clarity:
- Conciseness and focus – We shortened the manuscript by removing redundant descriptions and speculative content.
- Rare obstructive presentation – We added a dedicated paragraph in the Discussion emphasizing the unusual obstructive presentation (acute pulmonary edema) and placing it in the context of published literature.
- Differential diagnosis with mitral stenosis – The Discussion was expanded to stress the importance of distinguishing myxoma-related obstruction from rheumatic mitral stenosis, particularly relevant in our case with a diastolic murmur.
- Previous acute coronary syndrome – The speculative link to the myocardial infarction five years earlier was removed. This event is now presented only as medical history, clarifying that no mass was evident at that time and that the current presentation represents a new, independent finding.
- Histopathology – We reduced the number of histopathological images from three to two and simplified the text, retaining only essential information needed to confirm the diagnosis.
We believe these changes have improved the clarity and focus of the manuscript and that the revised version now reflects the reviewer’s valuable recommendations.

Reviewer 2 Report
Comments and Suggestions for Authors
1. The introduction is comprehensive, but somewhat verbose. A few suggestions can be shortened without losing scientific accuracy. For example, in lines 36–45, information about the frequency and localization of cardiac myxoma is repeated several times. A more concise structure would improve readability.
2. Lack of epidemiological context. Although the prevalence of cardiac myxomas is mentioned, the authors do not provide epidemiological data on the incidence rate in the population, age distribution, or gender differences. This would increase the clinical relevance of the entry.
3. Insufficient integration of the literature. The references provided (e.g. [1], [2]) are few and somewhat outdated. More recent reviews or large cohort studies of cardiac myxomas could supplement this information. At present, the references are mainly descriptive and do not contain a critical assessment of the state of the literature.
4. The clinical significance needs to be expanded. This section highlights the classic triad of symptoms (obstructive, embolic, constitutional), which is appropriate. However, the discussion would benefit from a quantitative assessment of risks (e.g., embolic complications occur in X% of cases; obstructive manifestations account for Y%). Without numerical context, the clinical significance remains unclear.
5. Abrupt transition to the case context. The introduction moves from general characteristics to “as shown in this case” (line 52), which prematurely mixes general information with the discussion of a specific case. A more standard structure would keep the introduction general, leaving the details of the specific case for the case description section.
6. The diagnostic approach requires a balanced assessment. The authors correctly emphasize echocardiography as the first-line method, but the discussion does not mention potential difficulties (e.g., small tumors may be missed; operator dependence; difficulty in differentiating a thrombus from a myxoma). Mentioning the limitations would have made the section more critical and balanced.
7. References to guidelines. The statement that surgical resection is recommended “regardless of symptom severity” (lines 66–68) is correct but requires an appropriate reference to a guideline (e.g., ESC or AHA guidelines). Currently, this statement risks appearing too general without reference to authoritative sources.
Comments on the Quality of English LanguageSome sentences are overly long and complex (e.g., lines 40–44). Shorter, more direct phrasing would improve clarity for international readers.
Author Response
We are grateful for the reviewer’s thoughtful and constructive comments, which have significantly improved the clarity and scientific rigor of our manuscript. Below we provide a detailed point-by-point response, with all modifications highlighted in green in the revised version.
- Introduction redundancy
We thank the reviewer for this valuable observation. The Introduction was revised to remove redundancies, and the information about frequency and localization of cardiac myxomas has been condensed into a single, concise sentence. - Epidemiological context
We thank the reviewer for this helpful suggestion. We have now added epidemiological information regarding incidence, typical age distribution, and gender predominance. These additions provide clearer clinical context. - Integration of recent literature
We appreciate this important remark. In the revised manuscript, we have integrated more recent references, including comprehensive reviews and a meta-analysis, to provide an updated and critical overview of the literature. These additions strengthen the scientific context of the manuscript. - Expansion of clinical significance
We thank the reviewer for this important suggestion. In the revised Introduction, we expanded the section on clinical presentation by including quantitative context for the different symptom clusters. This addition strengthens the clinical significance of the manuscript. - Abrupt transition to case context
We thank the reviewer for pointing this out. The Introduction has been revised to maintain a general focus without referring to the present case. The phrase linking directly to the case has been removed, and the sentence was reformulated to describe the rare presentation of acute pulmonary edema in general terms. - Balanced assessment of echocardiography
We appreciate this valuable observation. In the revised Introduction, we have added a statement highlighting the limitations of echocardiography, including possible diagnostic challenges. We also emphasized that advanced imaging modalities may be useful in selected cases. These changes provide a more balanced perspective. - References to guidelines
We thank the reviewer for this valuable remark. The statement regarding early surgical resection has been updated to include authoritative guideline references, which recommend prompt excision of cardiac myxomas irrespective of symptom severity. This correction provides the necessary authoritative support.

Round 2
Reviewer 1 Report
Comments and Suggestions for Authors
A significant improvement.
Still needs work on concision - would suggest word count limited to 2/3 of the current one.
Author Response
We thank the reviewer for the constructive feedback. In response, we clarified the results by directly linking imaging and intraoperative findings to the pathophysiology of obstruction, expanded figure legends, and revised the Case Presentation to emphasize the clinical impact of obstruction. The Discussion was carefully streamlined to reduce redundancy while preserving essential clinical and pathophysiological details; sections on growth kinetics, diagnostic ambiguity, and imaging modalities are now presented more concisely. Although we could not reduce the length to two-thirds without compromising important context, the revised version is substantially more focused and clear. All modifications are highlighted in blue.

Reviewer 2 Report
Comments and Suggestions for Authors
No
Author Response
We thank the reviewer for their positive assessment. In addition to maintaining the strengths of the original version, we have improved clarity and concision throughout, as shown in blue highlights.
